# Protective Effects of Astaxanthin on Ochratoxin A-Induced Liver Injury: Effects of Endoplasmic Reticulum Stress and Mitochondrial Fission–Fusion Balance

**DOI:** 10.3390/toxins16020068

**Published:** 2024-01-30

**Authors:** Yiting Zou, Shiyi Zhang, Jian Yang, Chen Qin, Bo Jin, Zhenyu Liang, Shuhua Yang, Lin Li, Miao Long

**Affiliations:** Key Laboratory of Zoonosis of Liaoning Province, College of Animal Science and Veterinary Medicine, Shenyang Agricultural University, Shenyang 110866, China; zouyiting2020@stu.syau.edu.cn (Y.Z.); zhangshiyizsy@hotmail.com (S.Z.); yangjian2020@stu.syau.edu.cn (J.Y.); qinchen2021@stu.syau.edu.cn (C.Q.); jin0213999@163.com (B.J.); liang4272023@163.com (Z.L.); longmiao@syau.edu.cn (M.L.)

**Keywords:** ochratoxin A, astaxanthin, endoplasmic reticulum, mitochondria, oxidative stress, broilers

## Abstract

Ochratoxin A (OTA), a common mycotoxin, can contaminate food and feed and is difficult to remove. Astaxanthin (ASTA), a natural antioxidant, can effectively protect against OTA-induced hepatotoxicity; however, its mechanism of action remains unclear. In the present study, we elucidate the protective effects of ASTA on the OTA-induced damage of the endoplasmic reticulum and mitochondria in broiler liver samples by serum biochemical analysis, antioxidant analysis, qRT-PCR, and Western blot analysis. ASTA inhibited the expressions of *ahr*, *pxr*, *car*, *cyp1a1*, *cyp1a5*, *cyp2c18*, *cyp2d6*, and *cyp3a9* genes, and significantly alleviated OTA-induced liver oxidative damage (SOD, GSH-Px, GSH, MDA). Furthermore, it inhibited OTA-activated endoplasmic reticulum stress genes and proteins (*grp94*, GRP78, *atf4*, ATF6, *perk*, *eif2α*, *ire1*, CHOP). ASTA alleviated OTA-induced mitochondrial dynamic imbalance, inhibited mitochondrial division (DRP1, *mff*), and promoted mitochondrial fusion (OPA1, MFN1, MFN2). In conclusion, ASTA can decrease OTA-induced oxidative damage, thereby alleviating endoplasmic reticulum stress and mitochondrial dynamic imbalance.

## 1. Introduction

The existing studies on *Aspergillus* and *Penicillium* report that ochratoxin A (OTA) exerts toxic effects on various animals, including hepatotoxicity, nephrotoxicity, teratogenicity, immunotoxicity, neurotoxicity, and genotoxicity [1], with species and sex differences [2]. In animals, the kidneys are the first organ damaged by OTA poisoning, followed by the liver, then the intestine, spleen, and pancreas [3]. OTA can exert toxic effects in vivo by generating reactive oxygen species (ROS), inducing oxidative stress and lipid peroxidation, and inhibiting the phenylalanine metabolism pathway. The liver is also the major organ for lipid and phenylalanine metabolism [4,5]. OTA can induce liver injury via oxidative stress. Oxidative stress-induced liver injury alters the expression of cytochrome enzymes (CYPs) [6], which are widely distributed in the endoplasmic reticulum (ER) and mitochondrial inner membrane. The nuclear receptor superfamily (NR) regulates the expression of CYPs. CYPs are an important source of hydrogen peroxide. Their overexpression leads to hydrogen peroxide overproduction, thereby promoting oxidative stress development [7].

As an intracellular organelle, the ER plays a role in lipid synthesis, drug metabolism, secretory and membrane protein synthesis, post-translational modifications, and intracellular calcium (Ca^2+^) regulation [8]. Endogenous or exogenous factors lead to ER stress. These factors disrupt internal ER homeostasis, resulting in the excessive accumulation of unfolded or misfolded proteins in the ER. The major factors mediating endoplasmic reticulum stress are protein kinase R-like endoplasmic reticulum kinase (PERK), eukaryotic initiation factor 2α (eIF2α), recombinant activating transcription factor 4 (ATF4), recombinant activating transcription factor 6 (ATF6), and inositol-requiring enzyme 1 (IRE1). Furthermore, glucose-regulatory protein 78 (GRP78) acts as a coactivator of ER stress. GRP78 activates the folding of the hydrophobic regions of proteins in a calcium-dependent manner, thereby increasing C/EBP homologous protein (CHOP) expression, which is located at the intersection of three important ER stress-related signaling pathways. When external factors, such as inflammation, toxins, and drugs, disrupt the ER, they can disrupt endoplasmic reticulum homeostasis in hepatocytes and cause endoplasmic reticulum stress, leading to hepatic lipid metabolism disorders and liver injury [9].

Mitochondria are vital organelles that regulate cellular energy production, Ca^2+^ and redox homeostasis, and apoptosis [10]. Dynamin-related protein 1 (DRP1) binds to mitochondrial fission factor (MFF) and mitochondrial dynamin (MiD49/MiD51), which are cohesion proteins, and primarily regulates mitochondrial division. It is recruited to the mitochondrial outer membrane via the cytoplasm [11]. After the recruitment process is complete, DRP1 divides through self-polymerization around the mitochondrial outer membrane [12]. The ER regulates mitochondrial fission [13]. Mitochondrial fusion is a vital process for maintaining damaged mitochondria and restoring organelle health. Optic atrophy protein-1 (OPA1) in the inner membrane and mitochondrial fusion proteins 1 and 2 (MFN1/2) in the outer membrane are the main components involved in this process [14]. OPA1 regulates endosomal fusion and maintains the structure of the mitochondrial cristae via hydrolysis; this generates a long isoform of water-soluble proteins anchored at the N-terminal end of the endosomal membrane and short isoforms of water-soluble proteins anchored to the interstitial space of the membrane [15]. MFN1/2 facilitates membrane tethering and subsequent outer membrane fusion [16]. When the mitochondria divide and fuse in an unbalanced manner, mitochondrial damage occurs, leading to disturbances in hepatic lipid metabolism and, subsequently, liver injury [17].

Astaxanthin (ASTA) is a ketocarotenoid that is widely found in various microorganisms and marine animals [18]. ASTA displays several biological benefits, including potent antioxidant, DNA repair, and anti-stress properties [19]. Recent studies have reported that ASTA can attenuate oxidative stress-induced damage via the mitochondrial pathway. Furthermore, it can alleviate ER stress in mouse liver by inhibiting the PERK–eIF2α–ATF4–CHOP pathway and decreasing ATF6, IRE1, and XBP-1 expressions [20,21].

OTA is abundant in cereal products and animal feeds and is highly harmful to farming and human health. Broilers are highly sensitive to OTA, which is highly hepatotoxic, and the ER and mitochondria are important organelles for maintaining a normal liver metabolic function. Therefore, studying the toxic effects of OTA on the ER and mitochondria in broiler liver is crucial. ASTA exhibits excellent antioxidant, anti-inflammatory, and antiapoptotic effects; however, its protective mechanism against OTA-induced damage to the ER and mitochondria in broiler liver is not clear. This study aims to investigate the toxic mechanism and protective effects of OTA on mitochondrial dynamics and ER stress injury in broiler liver.

## 2. Results and Analysis

### 2.1. Intervention Effect of ASTA on OTA-Induced Liver Injury in Broilers

To investigate OTA-induced liver injury in broilers and the repair effect of ASTA, changes in body weight were recorded and liver index ratios were measured. Figure 1A–C illustrate that OTA toxicity slows the growth, decreases the average body weight, and extremely significantly decreases the liver index of broilers (*p* < 0.01). After the ASTA intervention, the liver index highly significantly increased (*p* < 0.05), indicating that ASTA alleviated the OTA-induced liver injury. Next, to further determine the intervention effect of ASTA on OTA-induced liver injury in broilers, serum biochemical indexes were measured and liver pathological sections were observed. Compared with the control group, AST and ALT activities were significantly increased in the OTA group (*p* < 0.01) (Figure 1D,E). Furthermore, compared with the OTA group, AST and ALT activities were significantly decreased in the OTA + ASTA group (*p* < 0.01). Figure 1F presents the HE-stained liver sections and reveals that the liver tissues from the control and ASTA groups are dense and compact, whereas those from the OTA group are loose, with inflammatory cell infiltration and fatty degeneration. The OTA + ASTA group was significantly better than the OTA group. Our findings suggest that OTA severely damages broiler livers and that ASTA protects the liver from OTA toxicity.

### 2.2. Effects of ASTA on OTA-Induced Liver Nuclear Receptors, Cytochrome Enzymes, and Oxidative Stress Indexes of Broilers

Previous studies have reported that ASTA is a highly effective antioxidant, therefore, we hypothesize that ASTA alleviates liver damage by decreasing OTA-induced oxidative stress responses. NRs and CYPs are crucial for toxin metabolism; their sharp increase in expression can aggravate response increases in oxidative stress [6]. Qrt-PCR was performed to measure the expression levels of NRs (*ahr*, *pxr*, and *car*) and CYPs (*cyp1a1*, *cyp1a5*, *cyp2c18*, *cyp2d6*, and *cyp3a9*). As expected, OTA induced a sharp increase in the expressions of NRs and CYPs in the liver (*p* < 0.01) (Figure 2A–H). However, after ASTA treatment, this trend was reversed (*p* < 0.01). Additional experiments revealed that OTA decreased SOD and GSH-Px activities and GSH content in the liver (*p* < 0.01) and increased MDA content (*p* < 0.01). After ASTA treatment, SOD (*p* < 0.01) and GSH-Px (*p* < 0.01) activities and GSH content (*p* < 0.01) increased, whereas the MDA content decreased (*p* < 0.01) (Figure 2I–K). Collectively, these findings suggest that OTA can dramatically increase the expressions of NRs and CYPs in the liver of broilers, thereby aggravating oxidative stress responses, and that ASTA can decrease the expressions of NRs and CYPs, decreasing OTA-induced oxidative stress.

### 2.3. Observation of Broiler Liver Ultrastructure

The elevated expression of liver cytochrome enzymes induced by OTA can indirectly lead to an overexpression of ROS, eventually causing damage to the ER and mitochondria [22]. Therefore, we hypothesized that ASTA could mitigate the liver damage caused by OTA by safeguarding the ER and mitochondria. To verify this theory, we examined the ultrastructure of broiler livers through transmission electron microscopy. As illustrated in Figure 3, the ER exhibits signs of damage, the mitochondria are swollen and ruptured, the morphology of the bilayer membrane is obscured, and mitochondrial cristae are dissolved after OTA treatment. In contrast, after the ASTA intervention, the ER and mitochondria exhibited a normal morphology, the mitochondrial bilayer membrane was visible, and a few mitochondrial cristae were observed. These experimental results align with our hypothesis, indicating that the ASTA intervention mitigated structural damage to the ER and mitochondria caused by OTA, thus preserving the a normal liver function.

### 2.4. Effects of ASTA on Liver ER Stress-Related Genes and Proteins in OTA-Induced Broilers

The transmission electron microscopy results reveal that ASTA can alleviate OTA-induced liver ER injury, and the finding is corroborated by qRT-PCR and Western blot. As shown in Figure 4A–H, the qRT-PCR results demonstrate that OTA induces the overexpression of ER stress-related factors, including *grp78*, *grp94*, *atf4*, *atf6*, *perk*, *eIF2α*, *ire1*, and *chop* genes (*p* < 0.01). Following ASTA treatment, the expression of these genes associated with ER stress was significantly reduced (*p* < 0.01). Additionally, Western blot tests (Figure 4I–K) revealed that the ATF6, CHOP, and GRP78 protein levels were all significantly lower (*p* < 0.01), and the expressions of ATF6 and CHOP were consistent with the qRT-PCR results. These results suggest that ASTA mitigates the structural and functional damage to the ER induced by OTA via the ER stress pathway, thereby inhibiting OTA-induced liver injury.

### 2.5. Effects of ASTA on OTA-Induced Expressions of Mitochondrial Dynamics-Related Genes and Proteins in Broilers

The key to maintaining the normal morphology and function of mitochondria lies in preserving a delicate balance between mitochondrial division and fusion. The disruption of this balance can lead to mitochondrial damage, consequently triggering cell apoptosis, which harms the organism. To verify this, we performed qRT-PCR and Western blot analyses to assess the expressions of genes associated with mitochondrial division and fusion. As shown in Figure 5A–E, the qRT-PCR results indicate that OTA treatment significantly promotes mitochondrial division (*drp1*, *mff*) (*p* < 0.01), while blocking mitochondrial fusion (*opa1*, *mfn1*, *mfn2*) (*p* < 0.01). However, when ASTA was administered in combination with OTA, the impact of OTA on mitochondrial dynamics was reversed. The Western blot results (Figure 5F–I) corroborated the qRT-PCR findings. These results suggest that ASTA alleviates OTA-induced mitochondrial functional impairment in broilers, potentially mitigating liver damage by restoring the mitochondrial dynamic equilibrium, inhibiting mitochondrial division, and promoting mitochondrial fusion.

## 3. Discussion

In this investigation, we observe how ASTA shields the liver from OTA harm. Previous research revealed that OTA could disrupt the arrangement of mouse liver cells, leading to disorganization, vanishing hepatic sinusoids, and cell degeneration and necrosis [23]. Two important indicators of liver damage are ALT and AST, whose levels rapidly increase when hepatocytes undergo cell death. The liver organ coefficient, a commonly used measure of liver health in animals, significantly decreases due to OTA exposure [24]. Studies have found that ASTA can reduce liver tissue fibrosis and cell necrosis in mice with alcoholic liver fibrosis, resulting in an improved liver organ coefficient, and decreased ALT and AST levels [25]. In this experiment, broilers in the OTA group exhibited considerably lower body weight and liver organ coefficients, whereas their ALT and AST levels were significantly higher; furthermore, we observed inflammatory cell infiltration and fatty degeneration in the liver. After the combined treatment with ASTA, we observed a significant increase in the organ coefficient and body weight, along with a significant decrease in ALT and AST levels. There was no evidence of inflammatory infiltration or fatty degeneration in the liver, confirming that ASTA could alleviate OTA-induced liver injury in broilers.

When harmful substances enter the body, the expression of CYP enzymes increases due to NRs migrating from the nucleus to the cytoplasm, initiating a phase-I metabolic response [26]. During this phase, AHR primarily regulates the expressions of CYP1, CAR primarily regulates the expression of CYP2, and PXR primarily regulates the expression of CYP3 [27]. OTA can be metabolized by CYP enzymes and NRs, which play a significant role in phase-I metabolism [28]. The overexpression of CYP enzymes can lead to a decreased level of GSH and aggravated oxidative stress [29]. Key antioxidant enzymes include SOD and GSH-Px, with GSH being the primary antioxidant. MDA is the primary byproduct of unsaturated fatty acid degradation and serves as a crucial indicator of oxidative stress [30,31]. Meki and Hussin et al. reported that OTA significantly increased the lipid peroxide levels in the liver while significantly reducing the activities of SOD and GSH-Px and depleting GSH levels [32]. Recent research indicates that ASTA can reduce acute liver damage by enhancing hepatic SOD expression [33,34]. ASTA achieves this by inhibiting the activity of CYP enzymes, thereby reducing the supply of electrons required for their function [35]. In mouse models of Alzheimer’s disease, ASTA was found to bind to docosahexaenoic acid as esters, resulting in reduced MDA levels and mitigating oxidative stress [36]. We assessed the mRNA expression of liver NRs, such as *ahr*, *car*, and *pxr*, as well as the levels of cytochrome enzymes, including *cyp1a1*, *cyp1a5*, *cyp2c18*, *cyp2d6*, and *cyp3a9*. We found that the expression of these genes was significantly elevated in the OTA group. Moreover, SOD, GSH-Px, and GSH levels in the broiler liver were considerably reduced, whereas MDA concentrations were significantly increased. These experimental results confirm that OTA induces a phase-I metabolic response in the liver, resulting in increased expressions of NRs and cytochrome enzymes, ultimately exacerbating oxidative stress in the liver. However, ASTA intervention successfully reversed this trend. These findings suggest that ASTA effectively inhibits OTA-induced phase-I metabolic responses, thereby reducing liver injury resulting from oxidative stress.

The ER is a significant organelle with essential roles in calcium homeostasis, lipid biogenesis, and the synthesis and folding of transmembrane proteins [37]. An irreversible stress response in the ER triggers ER degradation, inhibits protein translation, disrupts calcium homeostasis, and induces cell apoptosis [38]. GRP94 has been established as a marker of ER stress [39]. When the ER was stressed in mouse peritoneal macrophages, the expressions of ER stress-related protein factors ATF6, IRE1, and PERK were significantly increased [40]. Similarly, ER stress occurred in porcine renal epithelial cells, and the protein levels of ER stress-related factors GRP78, ATF4, IRE1, PERK, eIf2α, and CHOP were significantly increased [40]. The research indicates that ASTA can reduce liver ER stress in mice by inhibiting the gene expression and protein levels of ER stress-related factors, such as ATF6, ATF4, IRE1, PERK, eIF2α, and CHOP [20]. In our study, the OTA group exhibited significantly higher mRNA and protein levels of ER stress-related factors, including GRP78, *grp94*, *atf4*, ATF6, *ire1*, *perk*, *eif2α*, and CHOP. This observation, combined with the swelling and rupture of the broiler liver ER, suggests that OTA induces irreversible stress in the broiler liver ER. These results support the hypothesis that the ER is among the target organelles affected by OTA. Following ASTA intervention, the expression of ER stress-related factors in the liver was significantly reduced, and ER swelling was mitigated. Furthermore, the frequency of ER stress indicators decreased. GSH plays a crucial role in safeguarding cells from oxidative stress and damage related to ER stress [41]. GSH interferes with the redox state of the ER during the formation of disulfide bonds, contributing to ER stress following oxidative stress [42,43]. Haynes CM et al. found that the ratio of GSH:GSSG was significantly lower under oxidative stress, ranging from 1:1 to 7:1. At the same time, GSH has a higher nucleophilicity than parental glutathione, so GSH has a strong scavenging activity against oxidants. In addition to playing a role in disulfide bond formation, GSH also serves as a redox buffer source under oxidative stress conditions. When the cells are subjected to oxidative stress, a decrease in GSH levels hinders the reduction in oxidative stress damage and the formation of disulfide bonds, exacerbating ER stress [44]. Therefore, we concluded that OTA caused oxidative stress in the livers of broilers, which led to the decrease in GSH levels, and the decrease in GSH levels led to the occurrence of ER stress. ASTA alleviates liver ER stress by reducing oxidative stress and increasing GSH levels. These findings suggest that ASTA can reduce ER stress induced by OTA through blocking the expression of ER stress-related proteins, thereby reversing the liver damage caused by OTA.

Mitochondria play essential roles in energy metabolism, maintaining cellular REDOX homeostasis, and regulating cell apoptosis. Preserving the structure and integrity of mitochondria is crucial for the regular functioning of cells [45]. To maintain mitochondrial morphology and function, a delicate balance between division and fusion must be maintained [46]. Proteins, such as MFN1, MFN2, and OPA1, primarily regulate mitochondrial fusion, which is essential for upholding the integrity of mitochondrial structure and function [47]. GSH, a key component of the mitochondrial antioxidant defense system, plays a critical role in mitochondrial division and fusion. However, mitochondria cannot produce glutathione by themselves; thus, the GSH content in the liver is crucial for mitochondrial division and fusion [48]. The REDOX potential of GSH regulates mitochondrial fusion, with GSH promoting mitochondrial fusion through an MFN-dependent mechanism and increasing OPA1 oligomerization [49]. In the liver, the loss of MFN2 leads to mitochondrial disruption and triggers ER stress. Mitochondrial division is primarily regulated by DRP1 and MFF, and excessive mitochondrial division reduces mitochondrial oxidative phosphorylation and ATP content, damaging mitochondrial structure and function [50]. Studies have revealed that, in pig intestinal epithelial cells, deoxynivalenol can upregulate the genetic expressions of mitochondrial division factors DRP1 and MFF, downregulate the expressions of two mitochondrial fusion proteins, MFN1 and MFN2, disrupt the balance of mitochondrial division and fusion, cause mitochondrial morphological and functional damage, and induce a mitochondrial dynamics imbalance [51]. T-2 toxins cause mitochondrial morphological and functional damage by upregulating the expression of the mitochondrial division factor DRP1 and downregulating the expression of the mitochondrial fusion proteins OPA1, MFN1, and MFN2 in human liver 7702 cells [52]. It has been reported that heat stress can significantly increase the DRP1 protein level in mice, leading to changes in the mitochondria (the mitochondrial ridge in the hypothalamus disappears). After ASTA intervention, the DRP1 protein level significantly decreased, and the mitochondrial ridge was visible [53]. Our findings indicate a significant decrease in the gene expressions and protein levels of OPA1, MFN1, and MFN2 in the liver of the OTA group, but a significant increase in the expressions of *mff* and DRP1. Furthermore, mitochondrial abnormalities, such as swelling, rupture, an unclear structure, biolysis, and mitochondrial crisis, were observed in the OTA group. Based on these findings and our previous antioxidant index results, we conclude that OTA reduces liver GSH levels, leading to oxidative stress in broilers. Subsequently, this disrupts mitochondrial fusion, accelerates mitochondrial division, results in mitochondrial dynamic imbalance, damages mitochondria, and impairs mitochondrial function. Therefore, our study suggested that mitochondria were also a target organelle affected by OTA. The ASTA intervention, on the other hand, accelerates mitochondrial fusion, slows down mitochondrial division, restores mitochondrial homeostasis, and reduces mitochondrial damage. Transmission electron microscopy revealed that the morphology and structure of liver mitochondria were restored to normal after ASTA treatment. Combined with the previously measured antioxidant indices, ASTA could elevate liver GSH levels by mitigating OTA-induced oxidative stress in broiler liver. This, in turn, promoted mitochondrial fusion, retarded mitochondrial division, reinstated mitochondrial homeostasis, mitigated mitochondrial damage, and alleviated OTA-induced liver injury.

Based on the findings above, it can be hypothesized that OTA induces ER stress and mitochondrial damage by promoting oxidative stress in the liver, ultimately resulting in liver injury. ASTA has the potential to protect against liver damage caused by OTA by alleviating oxidative stress, reducing ER stress, and mitigating mitochondrial damage in the liver.

## 4. Conclusions

In this study, OTA could induce the expressions of NRs and cytochrome enzymes in liver to increase sharply, resulting in oxidative stress in broiler liver, ER stress, and disrupted mitochondrial homeostasis. These effects damaged the morphology and function of liver ER and mitochondria, thereby causing liver injury. On the other hand, the ASTA intervention reduced the expressions of NRs and cytochrome enzymes induced by OTA, inhibited oxidative stress in the broiler liver, alleviated ER stress, restored the balance of mitochondrial extremely dynamics, and restored the morphology and function of the ER and mitochondria. Consequently, ASTA played a protective role in preserving liver health.

## 5. Materials and Methods

### 5.1. Design of Animal Experiments

All animal experiments were conducted strictly in compliance with the laws of the Chinese People’s Republic on the Planning of Animal Models. Furthermore, all processes were conducted according to the Ethics Board of Shenyang Agricultural University and approved laboratory veterinary medicine principles (No. 201806014). Sixty white-feathered broilers (mean weight: 49.9 g; age: 1 day old) were reared in a three-layer cage with a temperature of 30 (±5) °C and a relative humidity of 40%. OTA was purchased from LKT Labs (St. Paul, MN, USA) and dissolved in 0.1 mol/L of sodium bicarbonate (NaHCO_3_) at an intragastric concentration of 1.0 mg/kg once daily. *Haematococcus pluvialis* powder (with an ASTA content of 1.14%) was purchased from Yunnan Aier Occurrence Technology Co., Ltd. (Yunnan, China). The ASTA concentration in the feed was 100 mg/kg. Sixty broilers were randomly categorized into four groups: control (Control), challenge (OTA), antidote (ASTA), and combination (OTA + ASTA). The experimental period was 7 days. All broilers were fed with free feeding and drinking water. The OTA and OTA + ASTA groups were given 0.2 mL of OTA solution daily, whereas the control and ASTA groups were given 0.2 mL of NaHCO_3_ solution daily. The broilers in the ASTA and OTA + ASTA groups were fed a diet with a 100 mg/kg ASTA concentration and the source of ASTA was *Haematococcus pluvialis* powder (the for was administered immediately after its preparation). In contrast, the broilers in the control and OTA groups received a regular feed. The doses of OTA and ASTA were determined using the data reported in previous studies [7,54].

### 5.2. Determinations of Body Weight and Liver Index of Broilers

Each group was weighed daily before feeding. At 24 h after the last administration, the experimental chickens were weighed before being sacrificed. The livers were stripped and cleaned with PBS three times. Any residual normal saline on the liver surfaces was then removed using filter paper before they were weighed.
Liver index = (liver mass/body mass) × 100%

### 5.3. Determination of Serum Biochemical and Antioxidant Enzyme Indexes

Serum alanine aminotransferase (ALT) and aspartate aminotransferase (AST) were detected using kits purchased from the Jiancheng Bioengineering Institute (Nanjing, China). Pro-oxidant (MDA) and antioxidant (SOD, GSH-Px, GSH) were detected with a liver tissue homogenate using a kit purchased from Wuhan Eliret Biotechnology Co., Ltd. (Wuhan, China). The testing procedure was carried out according to the instructions provided by the company above.

### 5.4. Hematoxylin–Eosin (HE) Staining Analysis

The liver tissues were fixed in 4% paraformaldehyde, followed by embedding in paraffin and subsequent sectioning. After these steps, the paraffin sections were dewaxed with xylene and washed with a gradient ethanol solution, and rinsed with distilled water. Then, they were stained with hematoxylin. After differentiating by hydrochloric ethanol, soaking in tap water, and staining with eosin, the liver tissue sections were dehydrated with gradient alcohol and xylene before being sealed with neutral resin. We observed the tissues through a light microscope (Leica DM750 microscope, Leica, Beijing, China) and took pictures.

### 5.5. Transmission Electron Microscopy

Liver tissues were cut into 1 mm thick cubes and fixed with glutaraldehyde overnight. After rinsing with PBS three times, the tissue blocks were transferred to 1% osmic acid and fixed for 1 h. After rinsing with PBS, the tissue blocks were dehydrated with a propionic acid gradient, embedded with epoxy resin, sliced, and stained; a transmitted electron microscope (Zeiss LSM 510, Zeiss, Shanghai, China) was used to observe them.

### 5.6. RNA Extraction and Real-Time Fluorescence Quantitative PCR

RNA was extracted from broiler liver tissue using an RNA extraction kit (Nanjing Noviacan Biotechnology Co., Ltd., Nanjing, China). Then, the extracted RNA was reverse-transcribed with a qRT-PCR kit (Nanjing Noviacan Biotechnology Co., Ltd., Nanjing, China) to obtain cDNA. β-actin was used as an internal reference. All primers were produced by Shengong Bioengineering Co., Ltd. (Shanghai, China), and the primer sequences are shown in Appendix A. The qRT-PCR mixture comprised 10 μL of 2× SYBRGREEN MIX (Nanjing Noviacan Biotechnology Co., Ltd., Nanjing, China), 0.8 μL of upstream and downstream primers (10 μmol·L^−1^), 2.0 μL of the cDNA template, and 6.4 μL of ddH_2_O. The reaction conditions were as follows: predenaturation at 95 °C for 30 s and 40 cycles of denaturation at 95 °C for 5 s, annealing at 55 °C for 30 s, and an extension at 72 °C for 30 s. The mRNA expression was calculated using the 2^−△△Ct^ method.

### 5.7. Western Blot

A protein extraction kit (Beijing Solarbio Science & Technology Co., Ltd., Beijing, China) was utilized to extract the total protein from the liver tissue. In this process, 0.1 g of liver tissue was placed into a centrifuge tube, and 1 mL of cold cracking liquid (10 μL each of phosphatase inhibitor, protease inhibitor, and PMSF) was added. The mixture was then ground with a disposable grinding pestle until there was no obvious tissue mass, and it was left for 30 min. Subsequently, the mixture was centrifuged at 4 °C at 12,000× *g* for 30 min. The supernatant produced after centrifugation was the total liver protein. A 5 × loading buffer was added according to the volume of protein obtained, and the denaturation condition was 100 °C for 10 min.

The initial voltage was 90 V, and the voltage was adjusted to 130 V 1 h later. The 400 mA constant flow film was used for 40 min, and the 1 × protein-free rapid-sealing solution was used in a constant-temperature shaker for 15 min at room temperature. The sealed PVDF membrane was cleaned 4 times in 1 × TBST buffer solution for 8 min each time, and then placed in a primary antibody overnight at a controlled temperature of 4 °C. The secondary antibody was incubated at room temperature for 2 h. After cleaning it 4 times with the PBS buffer for 8 min each time, it was developed after cleaning it with TBST 4 times. The name, dilution ratio, and manufacturer of the antibody used are shown in Appendix A.

### 5.8. Statistical Analysis

All the data results in this experiment follow the format of mean ± standard deviation (X ± SD). To assess differences among the groups, one-way analysis of variance and Tukey’s multiple comparison tests were conducted. Primer Premier 5.0 software (PREMIER Biosoft International, San Francisco, CA, USA) was used to design the PCR primers. A statistical analysis was performed using IBM SPSS Statistics 25 software (SPSS Inc., Chicago, IL, USA). GraphPad Prism 8 Software (GraphPad Software, USA) was used to draw the graphs. A *p*-value of <0.05 was considered to denote a significant difference, whereas a *p*-value of <0.01 was considered to denote an extremely significant difference.

## Figures and Tables

**Figure 1 toxins-16-00068-f001:**
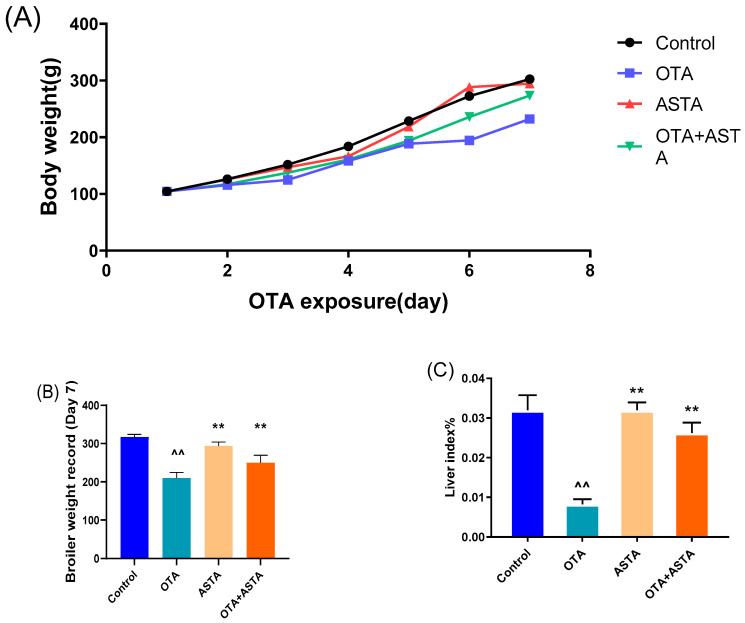
Effects of OTA on the body weight and liver injury of broilers. (**A**) Broiler body weight. (**B**) Broiler weight record. (**C**) Liver index. (**D**) ALT activity. (**E**) AST activity. (**F**) Pathological section of the livers of each group (200× and 400× magnifications). Pentagram: central vein; FG: fatty degeneration, black arrow; ici: inflammatory cell infiltration, red arrow. All values are expressed as AVG ± SD. **: Compared with the OTA group, *p* < 0.01. ^^: Compared to the control group, *p* < 0.01. ^: Compared to the control group, *p* < 0.05.

**Figure 2 toxins-16-00068-f002:**
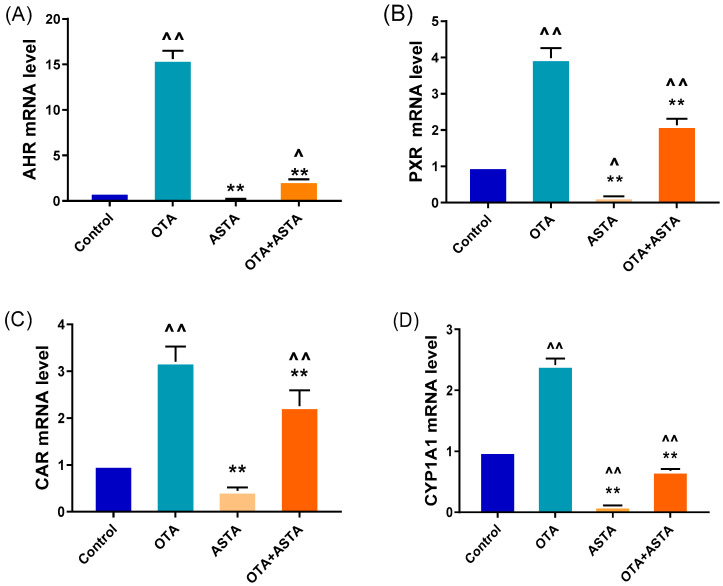
Effects of OTA on the expressions of nuclear receptor and cytochrome enzyme genes and oxidative stress indexes in broiler livers. (**A**–**H**) mRNA expressions of *ahr*, *pxr*, *car*, *cyp1a1*, *cyp1a5*, *cyp2c18*, *cyp2d6*, and *cyp3a9*; (**I**) SOD activity; (**J**) GSH-Px activity; (**K**) GSH level; (**L**) MDA level. Values are expressed as AVG ± SD. **: Compared with the OTA group, *p* < 0.01. ^^: Compared with the control group, *p* < 0.01. ^: Compared with the control group, *p* < 0.05.

**Figure 3 toxins-16-00068-f003:**
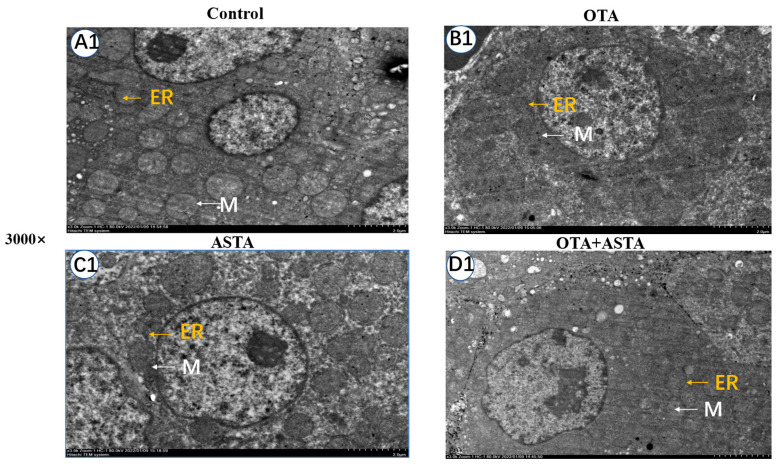
Ultrastructure of broiler liver cells under transmission electron microscopy. Magnifications of 3000 and 12,000 times. The 12,000-times-magnification image corresponds to the area within the yellow box in the 3000-times-magnification image. (**A1**–**D1**): Ultrastructure of liver cells at a 3000-fold visual field; (**A2**–**D2**): ultrastructure of liver cells at a 12,000-fold visual field. ER: endoplasmic reticulum; M: mitochondria; MC: mitochondrial ridge; MCM: mitochondrial bilayer membrane.

**Figure 4 toxins-16-00068-f004:**
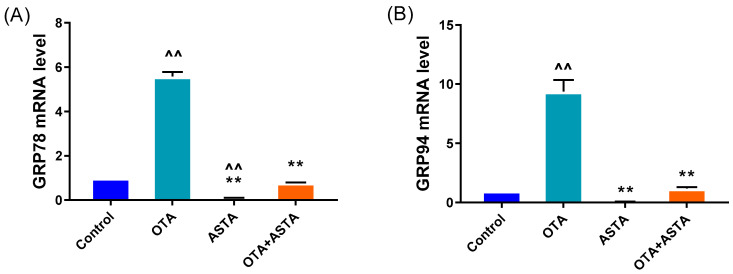
Effects of OTA on the gene and protein levels of liver endoplasmic reticulum stress factors in broilers. (**A**–**H**): *grp78*, *grp94*, *atf4*, *atf6*, *eif2α*, *ire1*, *perk*, and *chop* mRNA expressions. (**I**–**K**): CHOP, ATF6, and GRP78 protein levels. Values are expressed as AVG ± SD. **: Compared to the OTA group, *p* < 0.01. ^^: Compared to the control group, *p* < 0.01. ^: Compared to the control group, *p* < 0.05.

**Figure 5 toxins-16-00068-f005:**
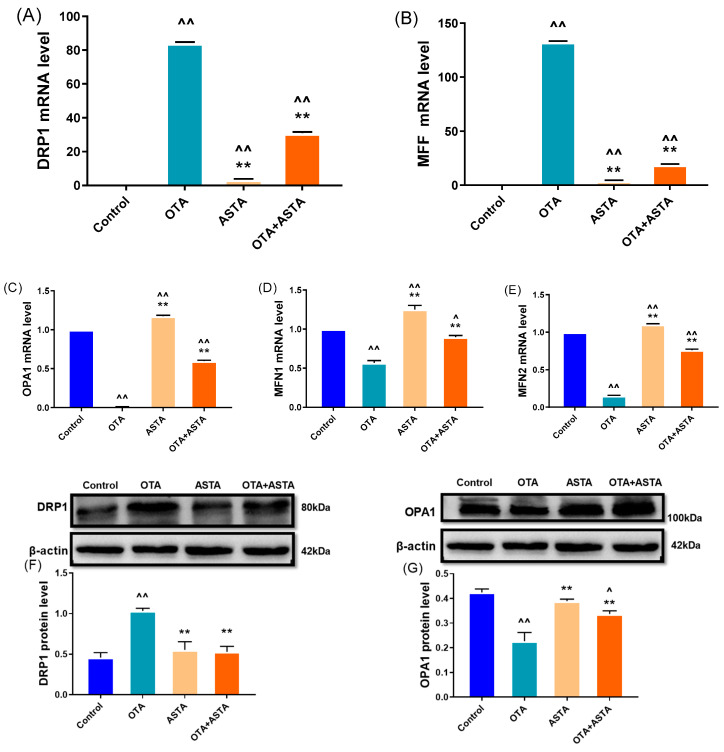
Effects of OTA on mitochondrial dynamic-related gene and protein levels in broiler liver. (**A**–**E**): *drp1*, *mff*, *opa1*, *mfn1*, and *mfn2* mRNA expressions. (**F**–**I**): DRP1, OPA1, MFN1, and MFN2 protein levels. Values are expressed as AVG ± SD. **: Compared to the OTA group, *p* < 0.01. ^^: Compared to the control group, *p* < 0.01. ^: Compared to the control group, *p* < 0.05.

## Data Availability

The article contains the information that was utilized to support the study’s conclusions.

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
