# Peer review of "Protective Effects of Astaxanthin on Ochratoxin A-Induced Liver Injury: Effects of Endoplasmic Reticulum Stress and Mitochondrial Fission–Fusion Balance"

_toxins, 2024, doi:10.3390/toxins16020068_

Round 1
Reviewer 1 Report
Comments and Suggestions for Authors
Toxins-2736813
Protective Effects of Astaxanthin on Ochratoxin A-induced Liver Injury: Effects of Endoplasmic Reticulum Stress and Mitochondrial Fission-Fusion Balance
The article is focused on the effect of a potent antioxidant as solution against OTA damage in broilers’ liver. Interestingly, it describes how OTA affect the structures, mainly to ER and mitochondria. This could be useful to know its impact in other organs such as kidney, due to it is a potent nephrotoxic. However, there are a lot of aspects to improve before being accepted for publication. Please, for future submissions, include the number of the lines because it is so much difficult to review without them.
Abstract: Remove a common fungal contaminant. Add a common mycotoxin.
Abstract: There is a lack of information about the methodology.
Abstract: the endoplasmic reticulum stress genes. Authors should clarify that ASTA overexpress or inhibit the gene expression that could be related to the mitochondrial dynamic imbalance etc. Because the study is only focused on gene expression.
Introduction: In animals, the kidneys are the first organ damaged….
In vivo in italics
It is not correct to say protein expression. The expression is from the genes that codify the proteins. For example, ATF6, IRE1, and XBP-1 expression.
2. Results
Figure 1A. Indicate the significant differences at least at day 7. Figures 1B, 1C and 1D: Make them higher and with the text at the same size. Revise the legend to include ASTA in the same line.
2.2. NRs and CYPs are crucial for toxin metabolism; their overexpression increases oxidative stress. Avoid overexpression because they are proteins. Change this throughout the manuscript.
The expression of genes that codify NRS (AHR….) and CYPs (CYP1A1…)
Figures 2I and 2K
Figure 2: Please, reorganize the figures because some times there are one in one line and other times there are two. Figure 2A and 2L are in bold.
Elevated expression of liver cytochrome enzymes induced… This paragraph is not well centrated.
Figure 3: Avoid the colour blue and green. It is difficult to read it. Maybe increase the size of the letters or change the colour.
2.4. The names of the genes should be written in lowercase and italics. The proteins in uppercase and without italics. Revise these aspects in all the manuscript.
Figure 4. Reorganize the figures to have the same structure and size. Which are the bands? They suddenly appear in the middle of the bars figures and they are not explain in the tittle of the figure. Remove it because the quality is not good and has no sense to include it inside figure 4. It could be supplementary material. The same for figure 5. Revise the name of the genes.
5.1. Design of animal experiments
5.2. Being sacrificied, not being killed. At 24 h after the last administration, the experimental chickens were weighed before being sacrified. The livers were stripped and cleaned with PBS three times.
5.6. Novazan, City, China
References: The references are not written using the style of the journal.
Comments on the Quality of English LanguageSome mistakes detected.
Author Response
Dear editor:
First of all, I wish you a merry Christmas.
On behalf of my co-authors, I thank you very much for giving us an opportunity to revise our manuscript, we appreciate the reviewer very much for the positive and constructive comments on our manuscript entitled “Protective Effects of Astaxanthin on Ochratoxin A-induced Liver Injury: Effects of Endoplasmic Reticulum Stress and Mitochondrial Fission-Fusion Balance.”.
We studied the reviewer’s comments carefully and had made revision according to the comments. In addition, we tried our best to revise our manuscript again and again in order to improve the quality of our manuscript. Revised contents in red were marked in revised manuscript (Please see revised version). We would like to appreciate for your kind consideration again. Our responses to the reviewer’s comments one by one are as following:
Reviewer #1: Protective Effects of Astaxanthin on Ochratoxin A-induced Liver Injury: Effects of Endoplasmic Reticulum Stress and Mitochondrial Fission-Fusion Balance
The article is focused on the effect of a potent antioxidant as solution against OTA damage in broilers’ liver. Interestingly, it describes how OTA affect the structures, mainly to ER and mitochondria. This could be useful to know its impact in other organs such as kidney, due to it is a potent nephrotoxic. However, there are a lot of aspects to improve before being accepted for publication. Please, for future submissions, include the number of the lines because it is so much difficult to review without them.
Response 1: Thank you for your review and suggestion. In response, we have added line numbers to the article for easy reading.
Abstract: Remove a common fungal contaminant. Add a common mycotoxin.
Response: Thank you for your review and suggestion. We have removed “common fungal contaminants” and replaced them with “common mycotoxins” in Line 6 of the abstract.
Abstract: There is a lack of information about the methodology.
Response: Thank you for your review and suggestion. We added Lines 8 through 10 in the summary“ In the present study, we elucidated the protective effects of ASTA on OTA-induced damage of the endoplasmic reticulum and mitochondria in broiler liver samples by serum biochemical analysis, antioxidant analysis, qRT-PCR, and Western blot analysis.”.
Abstract: the endoplasmic reticulum stress genes. Authors should clarify that ASTA overexpress or inhibit the gene expression that could be related to the mitochondrial dynamic imbalance etc. Because the study is only focused on gene expression.
Response: Thank you for your review and suggestion. We added “genes and proteins” in Lines 13-14. In this experiment, we not only tested the gene, but also tested the corresponding protein surface, so proteins were added.
Introduction: In animals, the kidneys are the first organ damaged….
Response: Thank you for your review and suggestion. We changed the “Broilers” to “animals” in Line 30.
In vivo in italics
Response: Thank you for your review and suggestion. We changed the “vivo” to “vivo” in Line 31.
It is not correct to say protein expression. The expression is from the genes that codify the proteins. For example, ATF6, IRE1, and XBP-1 expression.
Response: Thank you for your review and suggestion. We check the full text and changed all the “protein expression” to “protein level” such as Line 186、211、265.
- Results
Figure 1A. Indicate the significant differences at least at day 7. Figures 1B, 1C and 1D: Make them higher and with the text at the same size. Revise the legend to include ASTA in the same line.
Response: Thank you for your review and suggestion. We added a comparison of weight differences on day 7 and reformatted the images.
2.2. NRs and CYPs are crucial for toxin metabolism; their overexpression increases oxidative stress. Avoid overexpression because they are proteins. Change this throughout the manuscript.
Response: Thank you for your review and suggestion. We checked the entire manuscript and made all overexpression changes as follows:
Line11:“overexpression”to“expression”
Line123:“overexpression”to“sharp increase expression”
Line125-126:“overexpression”to“sharp increase in the expression”
Line131-132:“overexpression”to“dramatically increase the expressions”
Line133:“overexpression”to“expression”
Line254:“overexpression”to“increased expression”
Line340-341:“overexpression”to“expression……increase sharply”
The expression of genes that codify NRS (AHR….) and CYPs (CYP1A1…)
Response: Thank you for your review and suggestion. We went through the entire manuscript and changed all the genetic abbreviations to italic lowercase,changes as follows:
Line11-12:“AHR, PXR, CAR,CYP1A1, CYP1A5, CYP2C18, CYP2D6, CYP3A9” to “ahr, pxr, car, cyp1a1, cyp1a5, cyp2c18, cyp2d6, cyp3a9“
Line124-125:“NRs (AHR, PXR and CAR) and CYPs (CYP1A1, CYP1A5, CYP2C18, CYP2D6, and CYP3A9).” to “NRs (ahr, pxr, and car) and CYPs (cyp1a1, cyp1a5, cyp2c18, cyp2d6, and cyp3a9). “
Line142-143:“AHR, PXR, CAR,CYP1A1, CYP1A5, CYP2C18, CYP2D6, CYP3A9” to “ahr, pxr, car, cyp1a1, cyp1a5, cyp2c18, cyp2d6, cyp3a9“
Line248-250:AHR, PXR and CAR…CYP1A1, CYP1A5, CYP2C18, CYP2D6, and CYP3A9.” to “ahr, pxr, and car … cyp1a1, cyp1a5, cyp2c18, cyp2d6, and cyp3a9. “
Figures 2I and 2K Figure 2: Please, reorganize the figures because some times there are one in one line and other times there are two. Figure 2A and 2L are in bold.
Response: Thank you for your review and suggestion. We have reformatted all the graphics
Elevated expression of liver cytochrome enzymes induced… This paragraph is not well centrated.
Response: We have simplified this paragraph and changed Line151-158 to“As illustrated in Figure 3, the ER exhibited signs of damage, mitochondria were swollen and ruptured, the morphology of the bilayer membrane was obscured, and mitochondrial cristae were dissolved after OTA treatment. In contrast, after ASTA intervention, the ER and mitochondria exhibited normal morphology, the mitochondrial bilayer membrane was visible, and a few mitochondrial cristae were observed. These experimental results align with our hypothesis, indicating that ASTA intervention mitigated structural damage to the ER and mitochondria caused by OTA, thus preserving the normal liver function.“.
Figure 3: Avoid the colour blue and green. It is difficult to read it. Maybe increase the size of the letters or change the colour.
Response: Thank you for your review and suggestion. We changed the color and size of the symbols and reformatted them,the results are as follows:
2.4. The names of the genes should be written in lowercase and italics. The proteins in uppercase and without italics. Revise these aspects in all the manuscript.
Response: Thank you for your review and suggestion. We have reviewed the manuscript as a whole and made changes in accordance with your suggestions, which are as follows:
Line11-12:“AHR, PXR, CAR,CYP1A1, CYP1A5, CYP2C18, CYP2D6, CYP3A9” to “ahr, pxr, car, cyp1a1, cyp1a5, cyp2c18, cyp2d6, cyp3a9“
Line14:“GRP94,GRP78,ATF4,ATF6,PERK, eIF2α,IRE1,CHOP” to “grp94, GRP78, atf4, ATF6, perk, eif2α, ire1, CHOP “
Line15:“MFF” to “mff “
Line124-125:“NRs (AHR, PXR and CAR) and CYPs (CYP1A1, CYP1A5, CYP2C18, CYP2D6, and CYP3A9).” to “NRs (ahr, pxr, and car) and CYPs (cyp1a1, cyp1a5, cyp2c18, cyp2d6, and cyp3a9). “
Line142-143:“AHR, PXR, CAR,CYP1A1, CYP1A5, CYP2C18, CYP2D6, CYP3A9” to “ahr, pxr, car, cyp1a1, cyp1a5, cyp2c18, cyp2d6, cyp3a9“
Line171-172:“GRP94,GRP78,ATF4,ATF6,PERK, eIF2α,IRE1,and CHOP” to “grp94, GRP78, atf4, ATF6, perk, eif2α, ire1, and chop “
Line185:“GRP94,GRP78,ATF4,ATF6,PERK, eIF2α,IRE1,and CHOP” to “grp94, GRP78, atf4, ATF6, perk, eif2α, ire1, and chop “
Line197:“DRP1,MFF” to “drp1,mff “
Line198:“OPA1,MFN1MFN2” to “opa1,mfn1,mfn2 “
Line210:“DRP1,MFF OPA1,MFN1,and MFN2” to “drp1,mff,opa1,mfn1, and mfn2”
Line246-247:AHR, PXR and CAR…CYP1A1, CYP1A5, CYP2C18, CYP2D6, and CYP3A9.” to “ahr, pxr, and car … cyp1a1, cyp1a5, cyp2c18, cyp2d6, and cyp3a9.”
Line268:“GRP78,GRP94,ATF4,ATF6,PERK,eIF2α,IRE1,and CHOP” to “GRP78,grp94, atf4, ATF6, perk, eif2α, ire1, and CHOP”
Line317:“MFF“to”mff“
Figure 4. Reorganize the figures to have the same structure and size. Which are the bands? They suddenly appear in the middle of the bars figures and they are not explain in the tittle of the figure. Remove it because the quality is not good and has no sense to include it inside figure 4. It could be supplementary material. The same for figure 5. Revise the name of the genes.
Response: Thank you for your review and suggestion. We re-edited the protein bands and data analysis in Figure 4 and Figure 5 and indicated the corresponding bands with the corresponding group and size changes as follows:
5.1. Design of animal experiments
Response: Thank you for your review and suggestion. We have changed the title to“Design of animal experiments”
5.2. Being sacrificied, not being killed. At 24 h after the last administration, the experimental chickens were weighed before being sacrified. The livers were stripped and cleaned with PBS three times.
Response: Thank you for your review and suggestion. We have changed the corresponding part to “At 24 h after the last administration, the experimental chickens were weighed before being sacrified. The livers were stripped and cleaned with PBS three times.”
5.6. Novazan, City, China
Response: Thank you for your review and suggestion. Thanks for your suggestion, we have checked the company name and found that the previous name was filled in incorrectly. Now we have checked the full text and made changes as follows:
Line399-400:“Novazan“ to” Vazyme Biotech Co, China“
Line401:“Novozan” to “Vazyme Biotech Co., Ltd”
Line404:“Novozan” to “Vazyme Biotech Co., Ltd”
References: The references are not written using the style of the journal.
Response: Thanks to your suggestion, we have downloaded the reference format required by MDPI and made the full text changes as follows:
Line440-545:
- Zhu L, Zhang B, Dai Y, et al. 2017. A Review: Epigenetic Mechanism in Ochratoxin A Toxicity Studies[J]. Toxins, 9(4):
- Tao Y, Xie S, Xu F, et al. 2018. Ochratoxin A: Toxicity, oxidative stress and metabolism[J]. Food and chemical toxicology : an international journal published for the British Industrial Biological Research Association, 112(320-331.
- Stoev S D. 2010. Studies on carcinogenic and toxic effects of ochratoxin A in chicks[J]. Toxins, 2(4): 649-664.
- Abdel Halim A S, Rudayni H A, Chaudhary A A, et al. 2023. MicroRNAs: Small molecules with big impacts in liver injury[J]. Journal of cellular physiology, 238(1): 32-69.
- Tessari P, Vettore M, Millioni R, et al. 2010. Effect of liver cirrhosis on phenylalanine and tyrosine metabolism[J]. Current opinion in clinical nutrition and metabolic care, 13(1): 81-86.
- Xie Y, Hao H, Wang H, et al. 2014. Reversing effects of lignans on CCl4-induced hepatic CYP450 down regulation by attenuating oxidative stress[J]. Journal of ethnopharmacology, 155(1): 213-221.
- Li P, Li K, Zou C, et al. 2020. Selenium Yeast Alleviates Ochratoxin A-Induced Hepatotoxicity via Modulation of the PI3K/AKT and Nrf2/Keap1 Signaling Pathways in Chickens[J]. Toxins, 12(3):
- Anelli T Sitia R. 2008. Protein quality control in the early secretory pathway[J]. The EMBO journal, 27(2): 315-327.
- Lebeaupin C, Vallée D, Hazari Y, et al. 2018. Endoplasmic reticulum stress signalling and the pathogenesis of non-alcoholic fatty liver disease[J]. Journal of hepatology, 69(4): 927-947.
- Detmer S A Chan D C. 2007. Functions and dysfunctions of mitochondrial dynamics[J]. Nature reviews. Molecular cell biology, 8(11): 870-879.
- Palmer C S, Osellame L D, Laine D, et al. 2011. MiD49 and MiD51, new components of the mitochondrial fission machinery[J]. EMBO reports, 12(6): 565-573.4
- Kalia R, Wang R Y, Yusuf A, et al. 2018. Structural basis of mitochondrial receptor binding and constriction by DRP1[J]. Nature, 558(7710): 401-405.
- Friedman J R, Lackner L L, West M, et al. 2011. ER tubules mark sites of mitochondrial division[J]. Science (New York, N.Y.), 334(6054): 358-362.
- Yapa N M B, Lisnyak V, Reljic B, et al. 2021. Mitochondrial dynamics in health and disease[J]. FEBS letters, 595(8): 1184-1204.
- Otera H Mihara K. 2011. Molecular mechanisms and physiologic functions of mitochondrial dynamics[J]. Journal of biochemistry, 149(3): 241-251.
- Rojo M, Legros F, Chateau D, et al. 2002. Membrane topology and mitochondrial targeting of mitofusins, ubiquitous mammalian homologs of the transmembrane GTPase Fzo[J]. Journal of cell science, 115(Pt 8): 1663-1674.
- Lamanilao G G, Dogan M, Patel P S, et al. 2023. Key hepatoprotective roles of mitochondria in liver regeneration[J]. American journal of physiology. Gastrointestinal and liver physiology, 324(3): G207-g218.
- Jia Z, Song R, Xu Y, et al. 2022. Astaxanthin absorption modulated antioxidant enzyme activity and targeted specific metabolic pathways in rats[J]. Journal of the science of food and agriculture, 102(15): 7003-7016.2
- Chang M X Xiong F. 2020. Astaxanthin and its Effects in Inflammatory Responses and Inflammation-Associated Diseases: Recent Advances and Future Directions[J]. Molecules (Basel, Switzerland), 25(22):
- Wang W, Liu T, Liu Y, et al. 2021. Astaxanthin attenuates alcoholic cardiomyopathy via inhibition of endoplasmic reticulum stress-mediated cardiac apoptosis[J]. Toxicology and applied pharmacology, 412(115378.
- Wolf A M, Asoh S, Hiranuma H, et al. 2010. Astaxanthin protects mitochondrial redox state and functional integrity against oxidative stress[J]. The Journal of nutritional biochemistry, 21(5): 381-389.
- Wang Y, Cui J, Zheng G, et al. 2022. Ochratoxin A induces cytotoxicity through ROS-mediated endoplasmic reticulum stress pathway in human gastric epithelium cells[J]. Toxicology, 479(153309.
- Hassan R, Friebel A, Brackhagen L, et al. 2022. Hypoalbuminemia affects the spatio-temporal tissue distribution of ochratoxin A in liver and kidneys: consequences for organ toxicity[J]. Archives of toxicology, 96(11): 2967-2981.
- Li X N, Li H X, Yang T N, et al. 2020. Di-(2-ethylhexyl) phthalate induced developmental abnormalities of the ovary in quail (Coturnix japonica) via disruption of the hypothalamic-pituitary-ovarian axis[J]. The Science of the total environment, 741(140293.
- Wu Y C, Huang H H, Wu Y J, et al. 2019. Therapeutic and Protective Effects of Liposomal Encapsulation of Astaxanthin in Mice with Alcoholic Liver Fibrosis[J]. International journal of molecular sciences, 20(16):
- Humpage A R, Fontaine F, Froscio S, et al. 2005. Cylindrospermopsin genotoxicity and cytotoxicity: role of cytochrome P-450 and oxidative stress[J]. Journal of toxicology and environmental health. Part A, 68(9): 739-753.
- Ge J, Huang Y, Lv M, et al. 2022. Cadmium induced Fak -mediated anoikis activation in kidney via nuclear receptors (AHR/CAR/PXR)-mediated xenobiotic detoxification pathway[J]. Journal of inorganic biochemistry, 227(111682.
- Burkina V, Rasmussen M K, Pilipenko N, et al. 2017. Comparison of xenobiotic-metabolising human, porcine, rodent, and piscine cytochrome P450[J]. Toxicology, 375(10-27.
- Knockaert L, Fromenty B Robin M A. 2011. Mechanisms of mitochondrial targeting of cytochrome P450 2E1: physiopathological role in liver injury and obesity[J]. The FEBS journal, 278(22): 4252-4260.
- Crespo I, García-Mediavilla M V, Almar M, et al. 2008. Differential effects of dietary flavonoids on reactive oxygen and nitrogen species generation and changes in antioxidant enzyme expression induced by proinflammatory cytokines in Chang Liver cells[J]. Food and chemical toxicology : an international journal published for the British Industrial Biological Research Association, 46(5): 1555-1569.
- Del Rio D, Stewart A J Pellegrini N. 2005. A review of recent studies on malondialdehyde as toxic molecule and biological marker of oxidative stress[J]. Nutrition, metabolism, and cardiovascular diseases : NMCD, 15(4): 316-328.
- Meki A R Hussein A A. 2001. Melatonin reduces oxidative stress induced by ochratoxin A in rat liver and kidney[J]. Comparative biochemistry and physiology. Toxicology & pharmacology : CBP, 130(3): 305-313.
- Chen J T Kotani K. 2016. Astaxanthin as a Potential Protector of Liver Function: A Review[J]. Journal of clinical medicine research, 8(10): 701-704.
- Yang Y, Bae M, Kim B, et al. 2016. Astaxanthin prevents and reverses the activation of mouse primary hepatic stellate cells[J]. The Journal of nutritional biochemistry, 29(21-26.
- Ohno M, Darwish W S, Ikenaka Y, et al. 2011. Astaxanthin can alter CYP1A-dependent activities via two different mechanisms: induction of protein expression and inhibition of NADPH P450 reductase dependent electron transfer[J]. Food and chemical toxicology : an international journal published for the British Industrial Biological Research Association, 49(6): 1285-1291.
- Che H, Li Q, Zhang T, et al. 2018. Effects of Astaxanthin and Docosahexaenoic-Acid-Acylated Astaxanthin on Alzheimer's Disease in APP/PS1 Double-Transgenic Mice[J]. Journal of agricultural and food chemistry, 66(19): 4948-4957.
- Pagliassotti M J. 2012. Endoplasmic reticulum stress in nonalcoholic fatty liver disease[J]. Annual review of nutrition, 32(17-33.
- Hetz C. 2012. The unfolded protein response: controlling cell fate decisions under ER stress and beyond[J]. Nature reviews. Molecular cell biology, 13(2): 89-102.
- Eletto D, Dersh D Argon Y. 2010. GRP94 in ER quality control and stress responses[J]. Seminars in cell & developmental biology, 21(5): 479-485.
- Dong L, Tan C W, Feng P J, et al. 2021. Activation of TREM-1 induces endoplasmic reticulum stress through IRE-1α/XBP-1s pathway in murine macrophages[J]. Molecular immunology, 135(294-303.
- Rana S V S. 2020. Endoplasmic Reticulum Stress Induced by Toxic Elements-a Review of Recent Developments[J]. Biological trace element research, 196(1): 10-19.
- Zeeshan H M, Lee G H, Kim H R, et al. 2016. Endoplasmic Reticulum Stress and Associated ROS[J]. International journal of molecular sciences, 17(3): 327.
- Malhotra J D, Miao H, Zhang K, et al. 2008. Antioxidants reduce endoplasmic reticulum stress and improve protein secretion[J]. Proceedings of the National Academy of Sciences of the United States of America, 105(47): 18525-18530.
- Haynes C M, Titus E A Cooper A A. 2004. Degradation of misfolded proteins prevents ER-derived oxidative stress and cell death[J]. Molecular cell, 15(5): 767-776.
- Kim B Song Y S. 2016. Mitochondrial dynamics altered by oxidative stress in cancer[J]. Free radical research, 50(10): 1065-1070.
- Tilokani L, Nagashima S, Paupe V, et al. 2018. Mitochondrial dynamics: overview of molecular mechanisms[J]. Essays in biochemistry, 62(3): 341-360.
- Wada J Nakatsuka A. 2016. Mitochondrial Dynamics and Mitochondrial Dysfunction in Diabetes[J]. Acta medica Okayama, 70(3): 151-158.
- Patten D A, McGuirk S, Anilkumar U, et al. 2021. Altered mitochondrial fusion drives defensive glutathione synthesis in cells able to switch to glycolytic ATP production[J]. Biochimica et biophysica acta. Molecular cell research, 1868(1): 118854.
- Shutt T, Geoffrion M, Milne R, et al. 2012. The intracellular redox state is a core determinant of mitochondrial fusion[J]. EMBO reports, 13(10): 909-915.
- Coronado M, Fajardo G, Nguyen K, et al. 2018. Physiological Mitochondrial Fragmentation Is a Normal Cardiac Adaptation to Increased Energy Demand[J]. Circulation research, 122(2): 282-295.
- Zhang C, Zhang K F, Chen F J, et al. 2022. Deoxynivalenol triggers porcine intestinal tight junction disorder: Insights from mitochondrial dynamics and mitophagy[J]. Ecotoxicology and environmental safety, 248(114291.
- Yang J, Guo W, Wang J, et al. 2020. T-2 Toxin-Induced Oxidative Stress Leads to Imbalance of Mitochondrial Fission and Fusion to Activate Cellular Apoptosis in the Human Liver 7702 Cell Line[J]. Toxins, 12(1):
- Chen Y, Yu T Deuster P. 2021. Astaxanthin Protects Against Heat-induced Mitochondrial Alterations in Mouse Hypothalamus[J]. Neuroscience, 476(12-20.
- Tolba S A, Magnuson A D, Sun T, et al. 2020. Dietary supplemental microalgal astaxanthin modulates molecular profiles of stress, inflammation, and lipid metabolism in broiler chickens and laying hens under high ambient temperatures[J]. Poultry science, 99(10): 4853-4860.

Reviewer 2 Report
Comments and Suggestions for Authors
Contamination of food and particularly feed by fungal contaminants is an increasing problem. This study demonstrates that the adverse effects of Ochratoxin-A on the liver metabolism of broilers can be reduced significantly by Astraxanthin. The manuscript can be further improved by consideration of the points given below.
Results and Analysis
2.1 para 1 Ln 5 delete "extremely" substitute "highly"
Fig. 1 B-D Labelling and numbering are too small see Fig A for a more appropriate size..
Fig. 1 E Its difficult to see the structural details at 200X- could the 400X magnification be used instead and the Figure enlarged ?
Captions The font size of the captions throughout should be the same size as the main text for easier reading.
Fig 2 Could all the figures be put in pairs throughout to aid reading e. g. A + B, C+D, etc
Fig. 3 This needs to be enlarged. Why are a and B in parallel not C and D?
Figure 3 is too complex and should be 2 separate Figs. More detail required in the caption regarding the RNA results also in Fig. 5.
Captions and Figure labelling should be the same throughout.
3. Discussion para 1 Ln 11 "fa y" should be "fatty"?
Materials and Methods
P15 Could Table 1 be moved to an Appendix?
5.7 Western "bio ing" should be "blotting"?
Author Response
Dear editor:
First of all, I wish you a merry Christmas.
On behalf of my co-authors, I thank you very much for giving us an opportunity to revise our manuscript, we appreciate the reviewer very much for the positive and constructive comments on our manuscript entitled “Protective Effects of Astaxanthin on Ochratoxin A-induced Liver Injury: Effects of Endoplasmic Reticulum Stress and Mitochondrial Fission-Fusion Balance.”.
We studied the reviewer’s comments carefully and had made revision according to the comments. In addition, we tried our best to revise our manuscript again and again in order to improve the quality of our manuscript. Revised contents in red were marked in revised manuscript (Please see revised version). We would like to appreciate for your kind consideration again. Our responses to the reviewer’s comments one by one are as following:
Reviewer #2: Contamination of food and particularly feed by fungal contaminants is an increasing problem. This study demonstrates that the adverse effects of Ochratoxin-A on the liver metabolism of broilers can be reduced significantly by Astraxanthin. The manuscript can be further improved by consideration of the points given below.
Results and Analysis
2.1 para 1 Ln 5 delete "extremely" substitute "highly"
Response : Thank you for your review and suggestion, it has been modified as follow:
Linn 96:delete "extremely" substitute "highly"
Fig. 1 B-D Labelling and numbering are too small see Fig A for a more appropriate size.
Response: Thank you for your review and suggestion. We have changed all header sizes.
Fig. 1 E Its difficult to see the structural details at 200X- could the 400X magnification be used instead and the Figure enlarged ?
Response: Thank you for your review and suggestion. We have enlarged all the slices and changed them as follows:
Captions The font size of the captions throughout should be the same size as the main text for easier reading.
Response: Thank you for your review and suggestion. We have changed all header sizes.
Fig 2 Could all the figures be put in pairs throughout to aid reading e. g. A + B, C+D, etc
Response: Thank you for your review and suggestion. We rearranged the pictures.
Fig. 3 This needs to be enlarged. Why are a and B in parallel not C and D?
Response: Thank you for your review and suggestion. We have enlarged all the slices and changed them as follows:
Figure 3 is too complex and should be 2 separate Figs. More detail required in the caption regarding the RNA results also in Fig. 5.
Response: Thank you for your review and suggestion. We have enlarged all the slices.
Captions and Figure labelling should be the same throughout.
Response: Thank you for your review and suggestion. We made a change to the title.
- Discussion para 1 Ln 11 "fa y" should be "fatty"?
Response: Thank you for your review and suggestion. We checked it and changed it to“fatty”
Materials and Methods
P15 Could Table 1 be moved to an Appendix?
Response: Thank you for your review and suggestion. We have moved Table 1 to the Appendix
5.7 Western "bio ing" should be "blotting"?
Response: Thank you for your review and suggestion. We checked it and changed it to“blotting”

Reviewer 3 Report
Comments and Suggestions for Authors
Manuscript deals with a very important question of the protective effect of dietary supplements in mycotoxin poisoning.
Major concers are:
- What is "liver organ ratio" in chapter 2. Results and analysis. Its completely not clear if you mean by this liver index or liver ratio (AST/ALT), liver organ coefficient?
- p. 3 "... fatty degeneration" , it will probably be more appropriate to describe the observed changes in the tissue exactly
- abberviations "NR" and "T-SOD", should be explained. There are only 3 isoforms of SOD, no one is T-SOD. If you mean total SOD, it should be given
- "Overexpression of CYP enzymes can lead to decreased activity of GSH and aggravated oxidative stress [29]." - GSH is not enzyme it doesn´t decrease activity, only concentration.
- "GSH interferes with the REDOX state of the ER during the formation of disulfide bonds, contributing to ER stress following oxidative stress [42,43]. In addition to playing a role in disulfide bond formation, GSH also serves as a REDOX buffer source under oxidative stress. When the cells are subjected to oxidative stress, a decrease in GSH levels hinders the reduction of oxidative stress damage and the formation of disulfide bonds, exacerbating ER stress [44]." think a little better about the discussion here, or how to organize sentences as "the endoplasmic reticulum is far more oxidizing and the reported GSH:GSSG ratio is notably lower and ranges from 1:1 to 7:1" and "GSSH has higher nucleophilicity than parental glutathione, and therefore GSSH exhibits strong scavenging activities against oxidants". There is no reason to capitalize the redox state.
- It is very important to state the correct dosage, but it is not at all clear from the material and methodology section what doses of OTA were administered, how many times a day and how. On what basis was the dose of OTA chosen?
- ASTA concentration - again it is not clear why this dose was chosen. Is it a safe dose due to metabolism? How was the powder administered? How was it ensured that individuals consumed their intended dose?
Comments on the Quality of English LanguageIt is necessary to have the text checked by a native English speaker.
Author Response
Dear editor:
First of all, I wish you a merry Christmas.
On behalf of my co-authors, I thank you very much for giving us an opportunity to revise our manuscript, we appreciate the reviewer very much for the positive and constructive comments on our manuscript entitled “Protective Effects of Astaxanthin on Ochratoxin A-induced Liver Injury: Effects of Endoplasmic Reticulum Stress and Mitochondrial Fission-Fusion Balance.”.
We studied the reviewer’s comments carefully and had made revision according to the comments. In addition, we tried our best to revise our manuscript again and again in order to improve the quality of our manuscript. Revised contents in red were marked in revised manuscript (Please see revised version). We would like to appreciate for your kind consideration again. Our responses to the reviewer’s comments one by one are as following:
Reviewer #3:Manuscript deals with a very important question of the protective effect of dietary supplements in mycotoxin poisoning.
Major concers are:
- What is "liver organ ratio" in chapter 2. Results and analysis. Its completely not clear if you mean by this liver index or liver ratio (AST/ALT), liver organ coefficient?
Response: Thank you for your review and suggestion, it has been modified as follow:
Line95:"liver organ ratio" to "liver index"
- p. 3 "... fatty degeneration" , it will probably be more appropriate to describe the observed changes in the tissue exactly
Response: Thank you for your review and suggestion. Because we did HE staining observation of liver tissue, tissue description was not included.
- abberviations "NR" and "T-SOD", should be explained. There are only 3 isoforms of SOD, no one is T-SOD. If you mean total SOD, it should be given
Response: Thank you for your review and suggestion. NR is explained on line 37“The nuclear receptor superfamily (NRs)”. We checked the full text and changed all “T-SOD” to“ SOD”.
- "Overexpression of CYP enzymes can lead to decreased activity of GSH and aggravated oxidative stress [29]." - GSH is not enzyme it doesn´t decrease activity, only concentration.
Response: Thank you for your review and suggestion. We examined the full text and changed “the activity” to “the level”
- "GSH interferes with the REDOX state of the ER during the formation of disulfide bonds, contributing to ER stress following oxidative stress [42,43]. In addition to playing a role in disulfide bond formation, GSH also serves as a REDOX buffer source under oxidative stress. When the cells are subjected to oxidative stress, a decrease in GSH levels hinders the reduction of oxidative stress damage and the formation of disulfide bonds, exacerbating ER stress [44]." think a little better about the discussion here, or how to organize sentences as "the endoplasmic reticulum is far more oxidizing and the reported GSH:GSSG ratio is notably lower and ranges from 1:1 to 7:1" and "GSSH has higher nucleophilicity than parental glutathione, and therefore GSSH exhibits strong scavenging activities against oxidants". There is no reason to capitalize the redox state.
Response: Thank you for your review and suggestion. it has been modified as follow:
Line277-280:Haynes CM et al. found that the ratio of GSH:GSSG was significantly lower under oxidative stress, ranging from 1:1 to 7:1. At the same time, GSH has a higher nucleophilicity than parental glutathione, so GSH has a strong scavenging activity against oxidants.
- It is very important to state the correct dosage, but it is not at all clear from the material and methodology section what doses of OTA were administered, how many times a day and how. On what basis was the dose of OTA chosen?
Response: Thank you for your review and suggestion. it has been modified as follow:
Line357-359:“OTA was purchased from LKT Labs (St. Paul, Minnesota, USA) and dissolved in 0.1 mol/L sodium bicarbonate (NaHCO3) at an intragastric concentration of 1.0 mg/kg once daily.”
The dosage of OTA was determined by referring to previous research results and combining with our previous studies. Three doses of 0.5 mg/kg, 1.0 mg/kg and 2.0 mg/kg were used for gradient attack with the same duration of 7 days to screen the optimal dose for acute attack. The experimental results showed that both 1.0 mg/kg and 2.0 mg/kg could significantly increase the content of ALT and AST, which proved that they could cause serious damage to the liver of broilers. In the 2.0 mg/kg group, two broilers died during the modeling period, indicating that 2.0 mg/kg concentration was too high. Therefore, 1.0 mg/kg was selected as our modeling concentration, and the experimental results are listed as follows:
- ASTA concentration - again it is not clear why this dose was chosen. Is it a safe dose due to metabolism? How was the powder administered? How was it ensured that individuals consumed their intended dose?
Response: Thank you for your review and suggestion. it has been modified as follow:
We also fed different concentrations of ASTA to screen the optimal antioxidant concentration, which were 40 mg/kg, 60 mg/kg, 80 mg/kg and 100 mg/kg, respectively. After one week, the changes of MDA, SOD, GSH-Px and T-AOC were detected. The test results showed that 100mg/kg feed concentration was the best dose, so we selected 100 mg/kg ASTA concentration in the feed as the dosage.
|
Groups |
MDA(nmol/ml) |
SOD(U/ml) |
GSH-Px(U/ml) |
T-AOC(U/ml) |
||
|
Control |
8.52±0.63c |
1038.71±112.62a |
1965.69±139.06a |
5.62±0.58a |
||
|
ASTA 40 mg/kg |
6.67±0.54b |
1361.91±178.62b |
2043.12±161.01a |
9.61±0.81b |
||
|
ASTA 60 mg/kg |
5.33±0.31a |
1862.47±132.05d |
2612.79±125.03b |
13.44±2.48cd |
||
|
ASTA 80 mg/kg |
6.06±0.33a |
1556.71±61.82c |
2431.71±135.98b |
12.18±2.61c |
||
|
ASTA 100 mg/kg |
4.95±0.28a |
1962.22±156.04d |
3035.61±155.39c |
15.26±1.23d |
||

Round 2
Reviewer 1 Report
Comments and Suggestions for Authors
After revision, the manuscript is better organize now. I thank the authors for carrying out the suggested changes. However, in the case of the proteins, I think I have not been able to explain that proteins are not expressed, only genes are expressed. Proteins are produced in greater or lesser quantities/abundances but are not expressed. This quality is only for genes.
The references still not follow the style of this journal "Toxins", because each MDPI journal can have a different style. The style required and specified in the instructions for authors is:
Journal Articles:
1. Author 1, A.B.; Author 2, C.D. Title of the article. Abbreviated Journal Name Year, Volume, page range.
Minor editing required.
Author Response
Dear editor:
We would like to thank you again for your constructive comments on our manuscript. We have made corresponding explanations and revisions. Please see the attachment for details.
Yours sincerely,

Reviewer 3 Report
Comments and Suggestions for Authors
Authors improved manuscript according to recommendations. I have no further comments.
Author Response
Dear editor:
On behalf of my co-authors, we appreciate the reviewer very much for the positive and constructive comments on our manuscript entitled “Protective Effects of Astaxanthin on Ochratoxin A-induced Liver Injury: Effects of Endoplasmic Reticulum Stress and Mitochondrial Fission-Fusion Balance.”.